# Suspected Frostbite Injuries in Coypu (*Myocastor coypus*)

**DOI:** 10.3390/ani12202777

**Published:** 2022-10-14

**Authors:** Friederike Gethöffer, Katharina M. Gregor, Isabel Zdora, Peter Wohlsein, Franziska Schöttes, Ursula Siebert

**Affiliations:** 1Institute for Terrestrial and Aquatic Wildlife Research, University of Veterinary Medicine Hannover, Foundation, Bischofsholer Damm 15, 30173 Hannover, Germany; 2Department of Pathology, University of Veterinary Medicine Hannover, Foundation, Bünteweg 17, 30559 Hannover, Germany

**Keywords:** *Myocastor coypus*, frostbite, dermatitis, invasive alien species

## Abstract

**Simple Summary:**

In its indigenous South American habitats, the coypu (*Myocastor coypus*) is exposed to milder climatic conditions than in most European countries. Owing to the good adaptability of this invasive species, the coypu has spread to regions with other climate zones. After the last severe frost period in February 2021 in Lower Saxony, Germany, several coypus with uncommon skin injuries were harvested. In this period, ground temperatures below −10° Celsius were recorded for approximately two consecutive weeks. The chronological sequence led to the assumption of frostbite injuries. The sampled animals consisted of five female and five male individuals. Post mortem examination revealed ulcerative to necrotizing lesions located predominantly on the tail and/or limbs. The cause of the lesions could not be determined with certainty. However, considering the local weather conditions and the distribution of lesions, frostbite has to be considered as the most likely cause.

**Abstract:**

Native to South America, the coypu (*Myocastor coypus*) is an invasive alien species (IAS) of Union concern. It was introduced to Germany a hundred years ago and is considered established in all German federal states. Between January and February 2021, ground temperatures below −10° Celsius were recorded in Lower Saxony, Germany, for approximately two consecutive weeks. Five male and five female coypus, harvested between 23 February and 31 March 2021, received a *post-mortem* examination. Nutritional status was poor in six cases, moderate in three and good in one case. Pregnancy was observed in two females. In all the animals, lesions were predominantly found on the distal limbs (*n* = 7) and/or tail (*n* = 10), involving the skin and soft tissue with occasional exposure or loss of bones. The histological findings consisted of chronic, ulcerative to necrotizing dermatitis and occasional ulcerative-suppurative dermatitis, necrotizing myositis, thrombosis, granulation tissue, fibrosis and intralesional dystrophic mineralization. Intralesional bacteria were present in six and fungal spores in one animal. Determination of the exact cause was not possible; however, considering the local weather conditions and the distribution of lesions, frostbite injuries have to be considered as the most likely cause. The intralesional bacteria and fungal spores most likely represent secondary contaminants. Interestingly, lesions of this kind have not been reported in coypus in Germany so far. Therefore, frostbite should be considered as a potential cause of disease in coypus, warranting further investigation.

## 1. Introduction

The coypu *(Myocastor coypus)* is listed as an invasive alien species (IAS) of Union concern [1]. Native to South America, coypus were originally introduced to Europe for fur farming. Its first appearance in Germany was reported about a hundred years ago [2,3] and it is now considered established in all German states [1,4], as well as in several European member states [5,6]. Several models predict a habitat expansion for coypu, depending on freshwater availability [7] and the number of freezing days [8] or monthly temperatures [9]. Subjected to hunting law in most German federal states, hunting bags of coypus are still increasing, indicating an ongoing population growth [10]. Invasiveness of this species affects habitat degradation [11,12] and damages the infrastructure, such as dams and dikes [13]. The poor adaptability of coypus to low temperatures has been discussed repeatedly in the past, despite its widespread and invasive nature. Thus, previous studies report reduced reproductive rates due to cold winters in Europe [14,15,16], which might have subsided due to climate change [17]. The ability to reproduce year-round and the incidence of mortality due to low temperatures are of particular interest. Between January and February 2021, an extreme frost period occurred in Northern Germany for two consecutive weeks. Shortly thereafter, hunters observed abnormal, apathetic behavior in coypus, and coypus suffering from acral skin lesions were repeatedly harvested. It was hypothesized that these injuries might derive from frostbite. Therefore, coypus were subjected to a *post-mortem* examination, collected from local hunters of the Federal State of Lower Saxony to validate the assumption of frostbite.

## 2. Materials and Methods

### 2.1. Animals and Post-Mortem Examination

Five male and five female coypus (*n = 10*) with severe skin injuries were captured and shot in six different areas in Lower Saxony, Germany, between 23 February and 31 March 2021 according to legal hunting procedures [10] and EU Regulation 1143/2014. The animals were harvested in rural parts of the municipalities Dorstadt, Einbeck, Lehrte, Nortmoor, Uetze and Wietmarschen (Table 1).

The animals were stored at −20 °C until necropsy. Age was determined based on body weight as well as developmental and dental status [18]. Skin lesions were graded as follows: no macroscopic changes (grade 0), erosive to ulcerative changes including superficial corium (grade 1), ulcerative changes affecting subcutaneous tissue (grade 2), ulcerative to necrotizing changes extending to the bone including the disruption of joint capsules (grade 3) and ulcerative to necrotizing changes including acroosteolysis (grade 4).

Collected organ and tissue samples were fixed in 10% neutral buffered formalin and routinely embedded in paraffin wax, sectioned at 2 µm and stained with hematoxylin-eosin (HE). In addition, individual samples were stained with periodic-acid Schiff (PAS) and elastica van Gieson (EVG).

### 2.2. Meteorological Data

The evaluation of air temperature near the animal capture site was conducted for a time period of 49 days (01 January to 18 February 2021) using weather recording data from the Climate Data Center provided by the German Weather Service (Deutscher Wetterdienst, DWD, https://cdc.dwd.de/rest/metadata/station/xy, accessed in 19 January 2022). DWD stations were selected based on proximity to the animal capture sites with a maximum distance of approximately 50 km: Ahaus, Emden, Göttingen, Hannover and Wolfsburg (Figure 1). Two ground levels were used for recording the minimum air temperature, 5 cm above ground (TG) and 2 m above ground (TN), while the mean air temperature was determined 2 m above ground (TM).

## 3. Results

### 3.1. Post-Mortem Findings

Five male and five female coypus received a full *post-mortem* examination. At necropsy, all ten coypus were classified as adults. The nutritional status was poor in six cases (60%; 6/10), moderate in three (30%; 3/10) and good in one case (10%; 1/10; Table 1). Two out of five females (40%; 2/5) were pregnant with eight fetuses each (case # 3, 7, Table 1).

All the animals presented skin lesions on the tail (100%; 10/10), while seven coypus also had skin lesions on the limbs (70%; 7/10; Table 2). Skin lesions of grade 1 were seen in five animals (50%; 5/10; Figure 2A). Grade 2 skin lesions were noted in four animals (40%; 4/10; Figure 2B). Seven animals each had skin lesions of grade 3 (70%; 7/10; Figure 2C) and grade 4 (70%; 7/10; Figure 2D and Figure 3). In addition, four animals presented a clear demarcation of skin and subcutaneous tissue between viable and non-viable tissue (4/10; 40%).

In addition to the gross pathological findings described above, histopathology revealed ulcerative to suppurative dermatitis (10%; 1/10), necrotizing myositis (10%; 1/10) and intralesional dystrophic mineralization (20%; 2/10) (Figure 4A). Moreover, two animals showed granulation tissue (20%; 2/10) and one animal showed fibrosis (10%, 1/10), each with incipient birefringence of collagen fibers under polarized light. Two animals had intralesional arterial thrombosis associated with intraluminal proliferation of fibrous tissue, as visualized by EVG stain (Figure 4B) and recanalization (20%; 2/10). Intralesional bacteria were present in six (60%, 6/10) and fungal spores in one animal (10%; 1/10; PAS reaction: positive). Minor macroscopic and histopathological findings are listed in Appendix A.

### 3.2. Meteorological Data

The temperature was checked on different ground levels at five weather stations closest to the animal capture sites at DWD online (https://cdc.dwd.de accessed on 19 January 2022). At the beginning of 2021, there was a period of ten consecutive days of severe frost with temperatures below −10° Celsius, starting on 6 February, with slight differences in ranges between the sites (Appendix A).

## 4. Discussion

IAS, such as coypus, are considered one of the biggest threats for biodiversity worldwide and impose enormous economic costs in the EU [19]. Owing to their high adaptability, naturally occurring impediments or restrictions for their spread are rare. To date, little is known about the animals’ resistibility to low temperatures. In Great Britain, so called strong winters have been discussed as a reason for the death or low reproductive rate of coypus [20]. In addition, cold winters are reported to have aided the British eradication campaign in the 1980s [21,22]. Similar observations of seasonal deaths were made following cold winter events in Eastern Germany in the 1990s [23], affecting not only coypus, but also muskrats, another semiaquatic IAS native to North America [24,25]. On the contrary, an experimental design showed that, depending on vegetation cover and sheet-ice formation, the coypu can survive severe winter conditions unharmed [26].

In this study, all the coypus had severe tail skin lesions of grade ≥ 3 and showed lesions on the limbs in varying degrees of severity. The morphological findings observed in the ten coypus are consistent with previously described frostbite lesions in animals [27]. Extremities, such as digits or tail tips, are the most commonly affected; as seen here, with local cold injuries that may range from a white to bluish discoloration to blister formation, ulceration, necrosis and dry gangrene that can affect epidermis, dermis and its appendages, as well as subcutis, muscles, tendons and bones. Both extensive ulceration and necrosis could be observed in the cases examined, some of them with demarcation or acroosteolysis. In addition, affected animals may show vascular changes, which were evident in two animals in the form of arterial thrombosis. The severity of the lesions may vary depending on the duration and intensity of the cold exposure. This may also have contributed to the varying degrees of frostbite observed, as limb lesions varied widely among the coypus. However, it should be noted that due to the small sample size, no clear relationship between the severity of lesions and origin of animals and thus, lowest air temperature, could be established. Moreover, given the variable extent of skin injury, the pain experienced in insults without nerve involvement could have led to self-mutilation. Unfortunately, although there was no indication of any other causative disease, other exogenous and endogenous noxae that may have caused or contributed to the skin lesions cannot be excluded with certainty. In addition, it cannot be conclusively determined whether the bacteria detected intralesionally in six animals have an etiologic significance or represent secondary or *post-mortem* contaminants.

Since coypus were captured at different time points after the frost period, the correlation of detected skin lesions with the extreme air temperatures is challenging. For this reason, histologic criteria for wound age determination must be consulted for a more accurate assessment. Leukocyte infiltration starts within 48 h after frostbite and remains visible for about two weeks. Only two animals showed signs of inflammation in the musculature, although cellular details were no longer visible due to extensive necrosis and/or autolysis. Whereas necrosis is seen at the earliest one week after the insult, a clear demarcation between viable and non-viable tissue, as seen in four coypus, can only be observed after weeks to months. Thrombus formation, on the other hand, is seen approximately 1–2 weeks after the insult, which was visible in two individuals. However, in one animal, the thrombosis was already in a remodeling process with recanalization. Furthermore, granulation tissue was present in two animals, whereas fibrosis was seen in only one coypu. Both alterations are indicative of a reparative process that is detectable at the earliest one week and three weeks after insult, respectively. All three cases presented incipient birefringence of collagen fibers under polarized light, which can at the earliest be observed after 21 days, depending on the animal species and localization [28]. This corresponds to the time elapsed from the frost period until the animals’ death. Thus, given the chronicity of some of these skin lesions, a correlation with the frost period is a high probability.

Nine animals were in a moderate to poor nutritional status following the severe cold period at the beginning of 2021. The skin lesions noted in the tail and limbs most likely affected foraging. Harvested in rural areas with some of the coldest air temperatures recorded (Appendix A), lack of food as well as the animals’ reluctance to leave their underground burrows to forage, must also be considered as factors for the poor nutritional status. Regardless, no other causes, such as parasites or dental malocclusion, were identified that could be related to the poor nutritional condition.

Last, three animals showed multifocal lymphocytic-plasmacellular pneumonia of mild to moderate intensity (Appendix A). This alteration cannot be allocated to a specific etiology. Nevertheless, cold air can induce several adverse effects, such as bronchoconstriction, mucosal defects and leucocyte infiltration [29,30,31,32,33,34,35]. Therefore, a causal relationship between the development of the observed lymphocytic-plasmacellular infiltrates and the documented climatic conditions cannot be excluded. However, similar findings can also be observed with regard to various animate or inanimate noxae and could, e.g., represent sequelae of previous infectious diseases. Therefore, their exact etiology remains unclear. One animal (Appendix A) had low-grade, focal, granulomatous pneumonia with an intralesional fungal structure (adiaspore). Similar findings are frequently observed in association with adiaspiromycosis caused by *Emmonsia* spp. [36]. Due to the very limited extent of this alteration, neither clinical relevance nor association with low temperatures was attributed to this finding.

Interestingly, the Eurasian otter *(Lutra lutra),* inhabitant of similar habitats as invasive coypus, seems to possess insulating air layers even under the tails fur [37]; and subcutaneous fat layers have been documented for the Canadian otter *(Lontra Canadensis)* at the sides of the tail [38], confirming the variety in anatomical adaptions of aquatic mammals [39]. Moreover, behavioral adaptions of rodents were reported in context with low temperatures [40].

In summary, the results suggest that unfavorable weather conditions might lead not only to reduced fat reserves and consequent reproductive failure, but also to severe frostbite, which might result in the long-term death of the coypu.

## 5. Conclusions

Among vertebrate IAS, the semiaquatic coypu is considered established in Germany as well as in many European member states. To date, little is known about the resistance of these animals to low temperatures. This study provides the first evidence of severe skin injuries in coypus that could result from frost periods. Although low extreme temperatures, such as those experienced in early 2021, are becoming less frequent due to climate change, it would be of interest to investigate the coypu population in a larger sample for potential injuries in the future. This includes comparison of rural and urban habitats under similar weather conditions.

## Figures and Tables

**Figure 1 animals-12-02777-f001:**
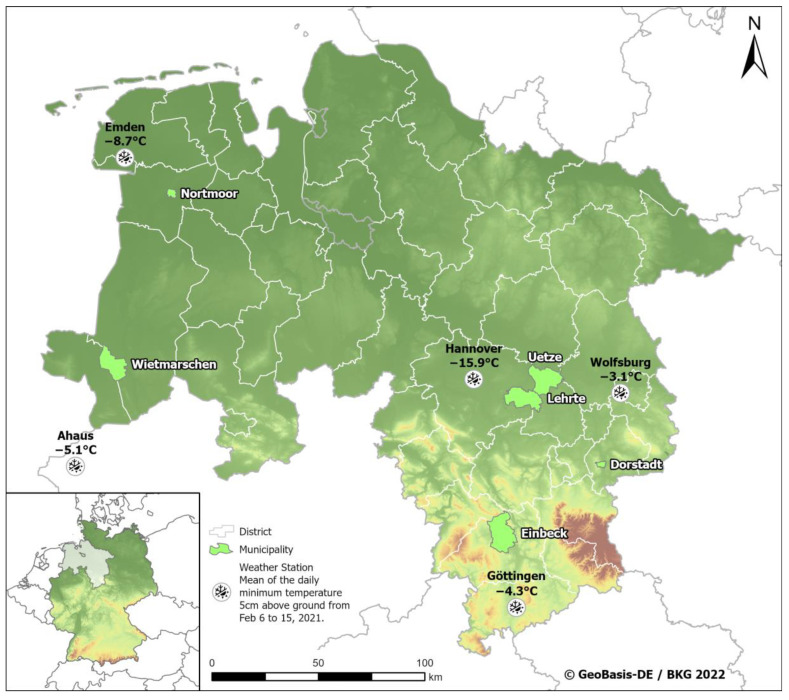
Map of Lower Saxony, Germany, with recorded weather data (stations in black bold font) and animal capture sites (municipalities shown in light green). Mean air temperature 5 cm above ground for the period from 6 until 15 February 2021, provided by the Climate Data Center of the German Weather Service for Ahaus, Emden, Göttingen, Hannover and Wolfsburg.

**Figure 2 animals-12-02777-f002:**
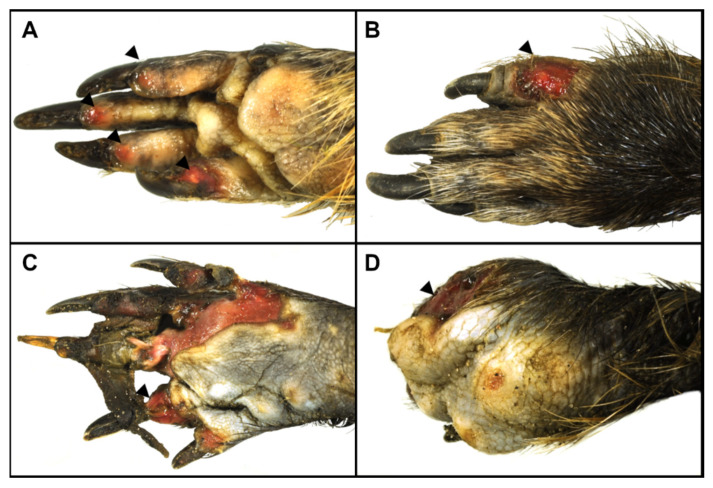
Skin lesions of the distal limbs in coypus (*Myocastor coypus*). Grade 1 skin lesion with erosive to ulcerative changes affecting the superficial corium of the toe pads (arrowheads; (**A**)). Grade 2 skin lesion characterized by ulcerative changes of the skin and subcutaneous tissue (arrowhead; (**B**)). Grade 3 skin lesion with ulcerative changes extending to the bone including disruption of joint capsules (arrowhead; (**C**)). Grade 4 skin lesion with necrosis and acroosteolysis of phalanges (arrowhead; (**D**)).

**Figure 3 animals-12-02777-f003:**
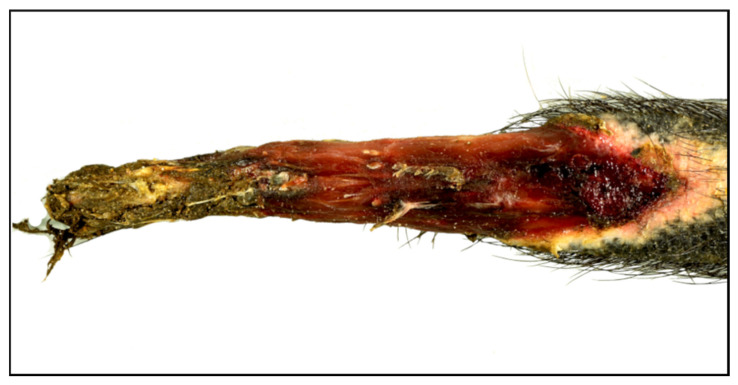
Skin lesion of the tail in coypus (*Myocastor coypus*). Tail with grade 4 lesion characterized by extensive ulceration and necrosis as well as loss of bone structures at the tip of the tail.

**Figure 4 animals-12-02777-f004:**
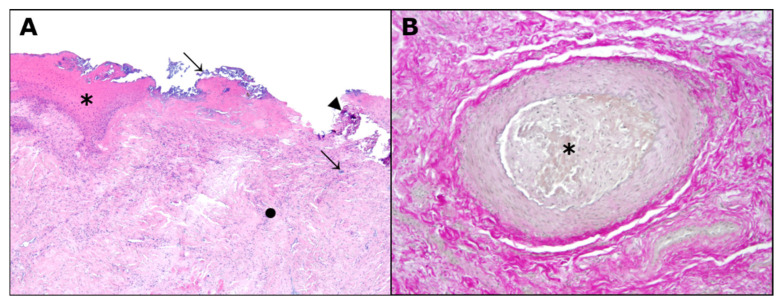
Cutaneous pathohistological findings in coypus. Severe extensive chronic ulcerative to suppurative dermatitis with superficial bacterial colonies (arrows) and intralesional dystrophic mineralization (arrowead). Note the reactive peripheral epidermal hyperplasia (asterisk) and the underlying granulation tissue (black dot), hematoxylin and eosin (**A**). Magnification 40×. Medium-sized cutaneous arterial vessel with thrombus (asterisk), elastica van Gieson (**B**). Magnification 100×.

**Table 1 animals-12-02777-t001:** Coypus (*Myocastor coypus*) examined in the present study, including animal number, date of capture, origin, sex, age and nutritional status. *: pregnant.

N°	Date of Capture	Origin	Sex	Age	Nutritional Status
1	2/23/2021	Nortmoor	Female	Adult	Moderate
2	3/26/2021	Wietmarschen	Female	Adult	Poor
3	3/30/2021	Wietmarschen	Female *	Adult	Moderate
4	3/31/2021	Wietmarschen	Male	Adult	Good
5	2/27/2021	Wietmarschen	Male	Adult	Poor
6	3/5/2021	Lehrte	Female	Adult	Poor
7	3/16/2021	Dorstadt	Female *	Adult	Poor
8	3/11/2021	Einbeck	Male	Adult	Poor
9	3/7/2021	Uetze	Male	Adult	Moderate
10	3/11/2021	Uetze	Male	Adult	Poor

**Table 2 animals-12-02777-t002:** Localization and grade of skin lesions in coypus (*Myocastor coypus*) examined.

N°	Localization and Grade of Skin Lesions				
	FL	FR	HL	HR	T
1	0	0	1	1	3
2	0	0	0	0	4
3	0	0	0	0	1, 4
4	0	0	0	0	4
5	2	2	2, 3	3	3, 4
6	1	1	1–3	1–3	3
7	0	0	1, 2	1	3, 4
8	2	2, 3	0	2	3
9	0	0	0	2, 3	2–4
10	1, 4	1, 4	3	1, 3	3

FL: left forelimb; FR: right forelimb; HL: left hindlimb; HR: right hind limb; T: tail; 0: skin without macroscopic changes; 1: skin with erosive to ulcerative changes including superficial corium; 2: skin with ulcerative changes including subcutaneous tissue; 3: skin with ulcerative to necrotizing changes extending to the bone including disruption of joint capsule; 4: skin with ulcerative to necrotizing changes including acroosteolysis.

## Data Availability

All the data supporting the findings of this study are included within the main document and are available upon reasonable request.

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
