# Peer review of "Suspected Frostbite Injuries in Coypu (Myocastor coypus)"

_animals, 2022, doi:10.3390/ani12202777_

Round 1

Reviewer 1 Report

This manuscript presents a case report of interest to wildlife veterinarias across Europe regarding what they should be expecting to find after the coldest periods of the year, specially in invasive species from warmest habitats.

The paper is well written, designed and presented. So, I only have a few suggestions for the authors to improve the quality of their case report. 

1) Adding a last paragraph of the introduction should clearly indicating the aims of your study. Were looking specifically for this kindly of injuries or it was a routine post-mortem exam or this animals? I know you mention the answer to these quesitons in other parts of your case report but I believe it should be clearly mentioned in a few words at the end of your introduction.

2) In one of your tables you present several lung lesions as well. Don't you think you should also comment on that in your discussion as you do with nutritional condition? Both things support your hypothesis that very cold temperatures are the cause of these injuries. Pneumonias and other respiratory conditions are often more seen in the winter, specially in animals not so well-adapted to cold temperatures. I believe a comment on that would enrich your discussion and hypothesis.

3) You can compare your findings with other species. Are there any other published reports on frostbite injuries in other species (including native)? Did you saw the same type of lesions in other speices during this same period? What are the differences found? A found a recent paper on an adaption to frostbite prevention in rodents: "tail‑belting". It may also complement your discussion. Stryjek, Rafal (2021) "A newly discovered behavior (‘tail‑belting’) among wild rodents in sub zero conditions".

Reviewer 2 Report

Overall, the manuscript is well written and includes interesting results. I would also include information from other temperate regions of the world that also have nutria (i.e., not only a European focus in terms of implications). For example, there are at least two (2) critical references (Ehrich 1962; Hilts et al. 2019) that the authors failed to include that discuss the implications of the low temperatures, number of freezing days, and formation of sheet ice relative to nutria range expansion, viability, and mortality. At a minimum, the authors should incorporate those references into their Introduction and Discussion. Anderson et al. (2022) may also be of interest to cite and incorporate into this paper.

Anderson et al. 2022. Confirming the broadscale eradication success of nutria (Myocastor coypus) from the Delmarva Peninsula, USA. Biological Invasions https://doi.org/10.1007/s10530-022-02855-x

Ehrich, S. 1962. Experiment on the adaptation of nutria to winter conditions. Journal of Mammalogy 43:418.

Hilts et al. 2019. Climate change and nutria range expansion in the eastern United States. Journal of Wildlife Management 83:591-598.
